# Multiple Antigenic Peptide-Based Vaccines Targeting *Ixodes ricinus* Neuropeptides Induce a Specific Antibody Response but Do Not Impact Tick Infestation

**DOI:** 10.3390/pathogens9110900

**Published:** 2020-10-28

**Authors:** Consuelo Almazán, Ladislav Šimo, Lisa Fourniol, Sabine Rakotobe, Jérémie Borneres, Martine Cote, Sandy Peltier, Jennifer Mayé, Nicolas Versillé, Jennifer Richardson, Sarah I. Bonnet

**Affiliations:** 1UMR BIPAR 0956, INRAE, National Veterinary School of Alfort, ANSES, Paris-Est University, 94700 Maisons-Alfort, France; c_almazan_g@hotmail.com (C.A.); ladislav.simo@vet-alfort.fr (L.Š.); fourniol-lisa@hotmail.fr (L.F.); sabine.rakotobe@anses.fr (S.R.); martine.cote94700@gmail.com (M.C.); 2SEPPIC Paris La Défense, 92250 La Garenne Colombes, France; jeremie.borneres@airliquide.com (J.B.); sandy.peltier@inserm.fr (S.P.); jennifer.maye@airliquide.com (J.M.); Nicolas.VERSILLE@airliquide.com (N.V.); 3UMR Virologie 1161, INRAE, National Veterinary School of Alfort, ANSES, Paris-Est University, 94700 Maisons-Alfort, France; jennifer.richardson@vet-alfort.fr

**Keywords:** multiple antigenic peptides, MIP, SIFamide, *Ixodes ricinus*, vaccines

## Abstract

Synthetic peptide vaccines were designed to target the neuropeptides innervating *Ixodes ricinus* salivary glands and hindgut and they were tested for their capacity to afford protective immunity against nymphs or larvae and *Anaplasma phagocytophilum*-infected nymph infestation, in mice and sheep, respectively. In both models, the assembly of SIFamide (SIFa) or myoinhibitory peptide (MIP) neuropeptides into multiple antigenic peptide constructs (MAPs) elicited a robust IgG antibody response following immunization. Nevertheless, no observable detrimental impact on nymphs was evidenced in mice, and, unfortunately, the number of engorged nymphs on sheep was insufficient for firm conclusions to be drawn, including for bacterial transmission. Regarding larvae, while vaccination of the sheep did not globally diminish tick feeding success or development, analyses of animals at the individual level revealed a negative correlation between anti-SIFa and MIP antibody levels and larva-to-nymph molting success for both antigens. Our results provide a proof of principle and precedent for the use of MAPs for the induction of immunity against tick peptide molecules. Although the present study did not provide the expected level of protection, it inaugurates a new strategy for protection against ticks based on the immunological targeting of key components of their nervous system.

## 1. Introduction

Among the 27 most important vector-borne diseases of humans listed by the World Health Organization (WHO), six are tick-borne diseases. These include Lyme disease, Crimean–Congo hemorrhagic fever, relapsing fever, rickettsial diseases (spotted fever and Q fever), tick-borne encephalitis, and tularemia [1]. *Ixodes ricinus* is the most abundant and widespread tick species in Europe and owing to both socio-economic and environmental changes exhibits ongoing changes in its distribution with respect to previously recognized endemic zones [2]. This tick species infests humans, as well as domestic and wild animals, and is the primary European arthropod vector of various human and animal pathogens, including members of *Borrelia*, *Anaplasma*, *Rickettsia*, and *Babesia* genera as well as tick-borne encephalitis virus [3].

An attractive strategy for controlling ticks is based on the identification of tick molecules of critical importance in the tick life cycle that could represent protective antigens on which anti-tick vaccines could be based [4]. This concept holds the promise of establishing an effective and environmentally benign control measure for these medically important arthropods along with the pathogens they transmit [5,6,7]. For the anti-tick vaccines developed to date, protection appears to be provided by vaccine-elicited antibodies that are ingurgitated with blood and that interact with the targeted antigen within the tick body, thereby interfering with its biological function.

In the past few decades several groups have evaluated the protective potential of diverse tick antigens playing roles in attachment, feeding, reproduction, development, bloodmeal digestion, water balance, vitellogenesis, and detoxification, as well as molecules that intervene at the tick–host interface in pathogen dissemination and/or transmission (see recent review by [8]). Although immunization with a number of tick antigens has led to the impairment of some of these processes and reduced tick viability or engorgement to some extent, to date, the only marketed anti-tick vaccine is the Gavac^TM^ (Heber Biotec S.A., Havana, Cuba) vaccine, targeting a *Rhipicephalus microplus* gut protein, Bm86 [9]. The effect of the recombinant Bm86 vaccine is based on reduction in tick infestation due to a diminished capacity of ticks to feed and—for the females—to subsequently reproduce [10]. This successful strategy underscored the vulnerability of the tick alimentary canal for establishing effective tick control. Other promising targets, prioritized by several laboratories, are the antigens found in tick salivary glands, as the secretory activities of this tissue play crucial roles at the tick–host interface and in pathogen transmission [11]. In fact, most tick-borne pathogens (TBP), following migration from the tick gut, mature/multiply in this organ prior to being secreted via saliva to the feeding cavity.

Neuropeptides are key signaling messengers which are synthetized from larger protein precursors in the cell body and subsequently processed post-translationally to form in many cases C-terminally amidated mature peptides. Mature neuropeptides are transported via axons to the secretory vesicles in the axon terminals serving as releasing sites close to target cell(s) or can be released directly from the neuroendocrine cell if acting as neurohormones [12]. Recently, two neuropeptides, SIFamide (SIFa) and myoinhibitory peptide (MIP), produced in specific cells in the tick central nervous system (synganglion), were identified as regulating the functions of both tick salivary glands and hindgut [13,14]. In salivary glands, they are suggested to regulate primary saliva expulsion from acini type II and III to the associated ducts, while in the hindgut an antagonistic effect on motility was documented. Although these studies have been performed mostly in the *Ixodes* genus, the physiological roles of SIFa/MIP are suggested to be common to the hard tick lineage [15,16]. Therefore, targeting the SIFa/MIP neuropeptides represents a rational approach for impacting two crucial processes in tick physiology, namely, saliva secretion and excretion of metabolic waste, as these two systems alternate during the extended feeding period of hard ticks.

Herein, we tested the effect of SIFa- and MIP-based multiple antigen peptide (MAP) vaccines on *I. ricinus* fitness during tick feeding, development and pathogen transmission. The impact of vaccination with both peptides was evaluated on infestation by nymphs in mice and by larvae and *Anaplasma phagocytophilum*-infected nymphs in sheep.

## 2. Results

### 2.1. Immunogenicity of Multiple Antigenic Peptides (MAPs) in Mice and Sheep

No clinical signs nor inflammation at the site of injection was observed after vaccination in any of the animals.

All the mice vaccinated with SIFa developed a specific IgG1 antibody response to the peptide that increased after the third immunization and remained at high levels until day 90 (Figure 1). For MIP, only 3 out of 5 mice developed a specific antibody response, which increased after successive immunizations and appeared to be stable until day 90 (Figure 1). Immunohistochemistry (IHC), using the antisera generated in mouse immunization experiments, revealed specific axons innervating the *I. ricinus* salivary gland acini for both SIFa and MIP (Figure 1).

Most of the sheep vaccinated with SIFa, MIP, or both peptides developed peptide-specific IgG responses, but this response was quite heterogeneous within each group of animals (Figure 2). With the exception of a single animal that had been vaccinated against both peptides but did not develop antibodies against either immunogen, the response increased in each case after successive immunizations, peaking at 45 dpi. The antibody titers in animals vaccinated with a combination of SIFa and MIP were similar to those in sheep vaccinated with either peptide alone, thus demonstrating the absence of either synergistic or competitive interaction between the immune responses elicited against the two peptides.

### 2.2. Efficacy of MAPs in Prevention of Tick Infestation and Development

Of 280 nymphs used to infest mice, 200 successfully engorged (71%), and, of these, 157 (78.5%) molted into the adult stage. Compared with the control group, the percentage of nymphs that fed to repletion was significantly higher in mice vaccinated with SIFa. Vaccination against MIP induced a significant decrease in engorged tick weight. On the contrary, vaccination led to a significant increase in the molting rate into adults for nymphs that fed on SIFa-vaccinated mice (Table 1). Finally, compared with the control, results showed a significant decrease in mean tick mortality rates in SIFa-vaccinated mice. However, when the results were analyzed at the individual level through ANOVA analysis, statistically significant differences between the experimental groups were not observed for any of the parameters (*p* > 0.05). No correlation between antibody levels and tick feeding and development parameters was demonstrated either for the group vaccinated with SIFa or with MIP (data not shown).

Out of a total of 20,924 larvae used to infest all the sheep included in the study, we collected a total of 15,459 engorged larvae for a general engorgement rate of 74%. Among these, 1602 molted into nymphs, corresponding to a global molting rate of 10%. Out of a total of 1152 nymphs used to infest sheep, only 27 engorged successfully (2%), from which 17 molted into adults (63%). The impact of the vaccination of sheep with SIFa, MIP or the combination of the two peptides on both larval and nymphal feeding, molting and mortality is shown in Table 2. For larvae, a statistically significant increase in the engorgement rate was observed in vaccinated sheep either with SIFa or MIP alone or with the combination of the two peptides when evaluated per group. The impact on the larval molting rate showed a similar trend for the group vaccinated with SIFa. On the contrary, the vaccination negatively affected the mortality rate of engorged larvae, with a global significant decrease in mean mortality for ticks that fed on SIFa- or MIP-vaccinated sheep as well as on sheep vaccinated with the combination of both peptides. However, at the individual level, the ANOVA analysis showed that none of the parameters displayed statistically significant differences between experimental groups (*p* > 0.05). The correlation between SIFa- or MIP-specific antibody levels in sheep vaccinated with SIFa or MIP alone or the combination of both peptides on larvae feeding, molting, and mortality is presented in Figure 3. A significant negative correlation was obtained between anti-SIFa antibody levels and tick mortality in sheep vaccinated with a combination of MIP and SIFa (r^2^ = 0.87, *p* = 0.023). A significant positive correlation was demonstrated between anti-MIP antibody levels and tick engorgement rates (r^2^ = 0.83, *p* = 0.04). Finally, a significant negative correlation was observed between anti-MIP antibody levels and molting rates of larvae that fed on MIP-vaccinated sheep (r^2^ = 0.82, *p* = 0.047).

Concerning the nymphs fed on vaccinated sheep, vaccination appeared to have a negative impact on tick engorgement and a positive one on tick mortality for nymphs that fed on SIFa-vaccinated sheep when compared with the control group. Nevertheless, no statistically significant difference was shown between groups when data were analyzed individually, and these results should be considered with caution because of the low total percentage of engorgement for nymphs on sheep, which is the reason why the correlation between antibody levels and impact on tick parameters was not evaluated here.

### 2.3. Impact of Vaccination on A. Phagocytophilum Transmission

Clinically speaking, only one sheep, belonging to the control group that had received only adjuvant, showed signs of *A. phagocytophilum* infection, including fever (Figure 4), lethargy, and inappetence five days after nymph infestation. In the other experimental animals, the temperature remained at normal values, and no other sign related to the infection was observed until the end of the experiment. The PCR analysis confirmed the infection of the sheep that showed clinical signs, with positive results at days 5 and 10 after nymph infestations, while PCR for the rest of the sheep was negative. Sequencing the PCR-amplified *msp4* gene demonstrated that the isolate corresponded both to the cultivated strain used to infect ticks and to *A. phagocytophilum* NV2Os, Gene Bank accession number CP015376.1. It should be noted that this sheep was the one on which the largest number of nymphs had engorged, with a total of 10 engorged nymphs collected out of 48 used versus a general average rate of engorgement of 2%. The sporadic infection of sheep with *A. phagocytophilum* was presumably related to the insufficient infestation rate for infected nymphs, which precluded evaluation of the impact of vaccination on pathogen transmission by ticks in this study.

## 3. Discussion

The anti-tick vaccine concept emerged in the 1940s [17], and since then several strategies for the development of vaccine candidates that interfere with the biological processes of ticks have been reported. However, the current study represents the first attempt to use small neuropeptide-based antigens to block specific neural processes of ticks. It has been suggested that any antigen not present in the host is a potential vaccine candidate if it encounters immunoglobulins and is associated with vital functions for the tick [10]. The two neuropeptides, SIFa and MIP, targeted in our study have been described in specific cells in tick synganglion as well as in axonal processes reaching both salivary glands and the hindgut [13,14]. In addition, different roles of these neuropeptides have been reported in insect where they are known to regulate diverse biological processes such as, for example, feeding, sexual behavior, molting, or ecdysis [18,19,20]. Therefore, we hypothesized that the generation of anti-tick immunity against these two neuropeptides might prevent ticks from successfully engorging and thus interfere with post-feeding development when feeding on immunized hosts.

The MAP-based immunization system has been utilized for generation of antibodies recognizing specific neuropeptides in the *Drosophila* central nervous system [21]. Herein, we took advantage of a universal T-helper epitope (PADRE) fused to either the SIFa or MIP peptides to induce antibody responses via the MAP system. Indeed, antibodies specific for SIFa or MIP were evidenced by ELISA after successive immunizations in both mice and sheep. Furthermore, PADRE itself did not elicit peptide-specific antibodies (data not shown), in keeping with its low immunogenicity described in previous reports [22]. The specificity of the generated antibodies was confirmed by IHC with antisera from immunized mice, which recognized axon terminals releasing SIFa and MIP in *I. ricinus* salivary gland acini. We did not evaluate in the present study whether antibodies acquired during a blood meal on immunized animals crossed the wall of the tick midgut and gained access to the haemocoel and hence the tick central nervous system, salivary gland, or hindgut, where they would be presumed to encounter endogenous SIFa and MIP and possibly interfere with their natural function. Rather, vaccine efficacy was directly evaluated through monitoring tick engorgement on immunized hosts and post-feeding tick parameters.

The peptides evaluated herein elicited a strong immune response in mice, as demonstrated by ELISA. However, as regards nymph infestation, vaccination with SIFa and MIP peptides did not afford protection, apart from modest diminution of the mean engorgement weight for the group immunized with MIP. Indeed, although immunogenicity is one of the key criteria used for the selection of tick vaccine candidates, the induction of antibodies does not necessarily ensure protection, as has also been observed by Coumou and co-workers who found that proteins homologous to Bm86 in *I. ricinus* elicited a strong immune response in rabbits but failed to confer protection against infestation with *I. ricinus* adults [23]. In addition, Prevot and coworkers found no efficacy against infestation with *I. ricinus* nymphs on mice immunized with Iris, an elastase inhibitor expressed in tick salivary glands, although a partial protection was found in nymphs and adults fed on immunized rabbits [24], suggesting that efficacy might vary depending on the mammalian species selected for evaluation.

In the present study, when results were evaluated in a target species (sheep), no protection against larvae was evidenced either, as revealed by comparison of the group average percentages of engorgement, molting success, or mortality between experimental groups. However, when these parameters were related to antibody levels at the individual level, a global negative correlation was evidenced between antibody levels against both peptides alone or in combination and molting success, which attains statistical significance for anti-MIP antibodies, suggesting that these antibodies may affect the molting of larvae into nymphs. Although this assumption needs to be further investigated, these data support the previous reports of MIP being involved in insect molting and ecdysis [18], while for SIFa these roles have not been reported. Taken together, targeting tick molting appears to be a promising tick control measure as ecological models of *Ixodes* ticks have shown that their population growth rate is highly sensitive to molting success [25]. Such a negative correlation has already been reported for molting of *I. ricinus* larvae to nymphs following immunization of rabbits with aquaporine [26] or the Q38 Subolisin/Akirin chimera vaccine [27]. With the nymph engorgement rate being only 2% on sheep, the result must be reproduced in order to evaluate the efficacy of vaccination of sheep on feeding and development of nymphs. Unfortunately, we do not have any explanation for the failure of *I. ricinus* nymph feeding on sheep in comparison to larvae, although the observation is in agreement with our recent observation [28].

## 4. Materials and Methods

### 4.1. Ethics Statement

All experiments with mice and sheep were performed according to the animal use protocol approved by the local ethics committee for animal experimentation, ComEth Anses/ENVA/UPEC, (Permit numbers 20150914113472401 and 2016092716395004) and in strict accordance with the recommendation of the European Guide for the Care and Use of Laboratory Animals [29].

### 4.2. Ticks

Pathogen-free *I. ricinus* larvae and nymphs were purchased from our tick rearing facility in ANSES (The French Agency for Food, Environmental and Occupational Health & Safety). Routinely, ticks were reared at 22 °C with 95% relative humidity and 12 h light/dark cycle and fed either using a membrane feeding method or on rabbits as previously described by Bonnet et al. [30] and Almazán et al. [31].

### 4.3. Synthesis of Multiple Antigenic Peptides

The vaccine candidates were synthetized as multiple antigenic peptides MAPs in which four copies of the same peptide were formulated on a lysine-based backbone (LifeTein, New Jersey, NJ, USA) (Figure 5). Each branch of the MAP candidate comprised a short *I. ricinus* non-amidated neuropeptide myoinhibitory peptide (MIP, ASDWNRLSGMW) [13], or SIFa, AYRKPPFNGSIF) [13] fused to a universal T helper cell epitope, the Pan DR epitope peptide (PADRE, AKFVAAWTLKAAA) [32]. The individual branches were chemically ligated to the lysine core. Masses of the synthetized MAPs were checked by Matrix Assisted Laser Desorption Ionization—Time of Flight (MALDI/TOF) system. Amidated linear forms of SIFa and MIP neuropeptides as well as PADRE were synthetized with >80% purity (LifeTein, New Jersey, NJ, USA).

### 4.4. Vaccine Formulation

Lyophilized MAPs were dissolved in dimethyl sulfoxide (DMSO) (Sigma-Aldrich, Darmstadt, Germany), at a concentration of 10 mg/mL and kept at −20 °C until use. Peptides were adjuvanted in Montanide™ ISA 201 VG (Seppic, La Garenne Colombes, France) 24 h before immunization and kept at 4 °C until use. Saline buffer (phosphate-buffered saline—PBS) was also adjuvanted in Montanide™ ISA 201 VG and used as the control. For vaccination 100 μL (containing 10 µg of MAPs) and 1 mL (containing 50 µg of MAPs)/dose were administrated per mice and sheep, respectively.

### 4.5. Immunization of Mice and Tick Infestation

Three groups of female C57BL/6 mice, at 8 weeks of age, were used. Two groups of 5 mice were vaccinated with either SIFa or MIP, while a control group of 4 mice received only adjuvant mixed with PBS (one mouse of the control group having died unexpectedly). All the mice were inoculated subcutaneously in the abdomen at days 0, 14, and 28 using 1 mL syringes fitted with 27G needles. Fourteen days after the last immunization (Day 42), each mouse was infested with 20 *I. ricinus* nymphs deposited in plastic capsules glued to the mouse’s back as previously described [33]. Engorged ticks were collected daily after 3 days of infestation until day 7 when all the ticks dropped off. Ticks were weighed and kept at 22 °C and >80% humidity. The tick nymphs that successfully molted to the adult stage were collected and counted after 109 days of incubation. Tick engorgement (No of engorged nymphs/No of nymphs used for infestation), mortality (No of dead nymphs at day 109/total No engorged nymphs), feeding (mean weight after feeding/engorged nymph), and molting (No adults at day 109/No engorged nymphs) were evaluated and compared between vaccinated mice and controls by student’s *t*-test with unequal variance (*p* = 0.05), *χ*^2^-test (*p* = 0.05) and one-way analysis of variance ANOVA (*p* < 0.05). Two-tailed Pearson’s correlation analysis was conducted in Microsoft Excel (Version 16.16.19; *p* < 0.05) to compare the effect of vaccination on engorgement, mortality, feeding, and molting after feeding on vaccinated or control mice with antibody titers at day 42, when tick infestation was performed.

### 4.6. Whole-Mount Immunohistochemistry

To confirm the specificity of antibodies against neuropeptides generated in C57BL/6 mice, we performed immunohistochemistry (IHC) on unfed *I. ricinus* salivary glands as both SIFa and MIP are known to be co-expressed in the axonal projections reaching this tissue [13]. We followed a protocol that had previously been established for tick-tissue IHC [13,34]. Briefly, salivary glands were dissected in ice-cold phosphate buffer from unfed *I. ricinus* adults and fixed with Bouin’s solution for 2 h at room temperature (RT), then washed with PBS + 0.5% Triton X-100 (PBST). Salivary glands were incubated for 3 days at 4 °C with antisera (diluted 1:300 in PBST) obtained from the mouse that showed the highest titer of antibodies on day 42 after first immunization. For the negative controls, pre-immune sera diluted 1:300 from the same mouse was used. After washing with PBST, the specimens were incubated overnight at 4°C with Alexa 488-conjugated goat anti-mouse secondary antibody (Life technologies, Carlsbad, CA, USA) diluted at 1:1000. Samples were mounted in Prolong^TM^ (Thermo Fisher Scientific, Waltham, MA, USA) antifade diamond mounting medium containing DAPI (Life Technologies, Carlsbad, CA, USA) and analyzed by inverted confocal microscopy using a Zeiss LSM 700 instrument. Image adjustment was performed in Adobe Photoshop CS6 (Adobe System Incorporate, San Francisco, CA, USA, 2012).

### 4.7. Immunization of Sheep and Tick Infestation

The vaccination and tick infestation were performed at the Platform of Infectiology, Val de Loire, INRAE Nouzilly, France. Four groups of six 7-month-old male sheep (PreAlps breed) were intramuscularly immunized as previously reported by Hope et al. [35], with three doses of SIFa, MIP, or the combination of SIFa and MIP, or adjuvant mix with PBS for the control group. Each animal was vaccinated using 5 mL syringes fitted with 16G needles at days 0, 15, and 30. Fifteen days after the last immunization, all of the animals were moved to a biosafety level 2 enclosure where the backs of the sheep were shaved, and cotton bags placed on both sides of the back as previously described [28]. After 24 h, the sheep were infested with approximately 1000 larvae and 24 h later with 48 nymphs that had been engorged as larvae on a sheep infected with *A. phagocytophilum* (NV2Os strain), resulting in an estimated infection rate in nymphs of 42%, as previously described [28]. After 24 h of infestation, all the unattached ticks were removed from the cotton bags and counted. Fully engorged ticks that dropped off into the cells were collected two times per day over 3 days of feeding for larvae and 4 days of feeding for nymphs. After every collection, ticks were washed, counted, and incubated at 22 °C and 80% of humidity until molting. The tick larvae and nymphs that successfully molted to nymph and adult stages, respectively, were collected and counted after 90 days of incubation.

For both larvae and nymphs, tick engorgement (No of engorged tick/No of tick used for infestation), mortality (No of dead tick at day 90/total No engorged-attached tick), and molting (No nymphs-adults at day 90/No alive engorged larvae-nymphs at day 90) were evaluated and compared between vaccinated sheep and controls by *χ*^2^-test (*p* = 0.05) and one-way analysis of variance ANOVA (*p* < 0.05). Two-tailed Pearson’s correlation analysis of the effect of vaccination on tick engorgement, mortality, and molting, after feeding on vaccinated or control sheep, with antibody titers at day 45, was conducted in Microsoft Excel (Version 16.16.19; *p* < 0.05).

### 4.8. Evaluation of Vaccine Innocuity

The innocuity of the vaccination was evaluated by observation of clinical signs and any local reaction at the site of the vaccine injection, changes in color and presence of inflammation were monitored every 24 h during three days after each immunization in both mice and sheep [36]. In sheep, any rise in temperature was monitored throughout the entire experiment using an intraruminal thermobolus system (Medria, Chateaugiron, France) that had been orally administered to each sheep seven days before the first vaccination. The temperature was recorded per hour, and the daily average was calculated and plotted against each day, starting one day before tick infestation (day 45), until day 54.

### 4.9. Follow-Up of A. phagocytophilum Infection in Sheep

Clinical signs, with special attention on the increase of temperature, were monitored in all sheep as described above. For *A. phagocytophilum* detection by PCR, blood samples were collected from the jugular vein of each sheep starting 1 day before infestation with *A. phagocytophilum infected I. ricinus* nymphs: then at 5, 10, and 15 dpi. Blood was *drawn* in 10 mL EDTA vacutainer tubes, and 200 µl used to isolate DNA by the NucleoSpin kit, according to the manufacturer’s instructions (Macherey-Nagel, Düren, Germany). PCRs were performed with primers msp4-F (5′-CCTTGGCTGCAGCACCACCTG-3′), and msp4-R (5′-TGCTGTGGGTCGTGACGCG-3′) for *A. phagocytophilum msp4* gene detection [37]. The reactions were performed in a final volume of 20 µl using the Takara Ex Taq system (Bio Europe, *Saint**-**Germain-en-Laye,* France). DNA from *A. phagocytophilum* infected ISE6 tick cells and nuclease free water were used as positive and negative controls, respectively. PCR products were visualized by 2% agarose gel electrophoresis. Amplicons were purified by using the PCR Clean-Up kit (Macherey-Nagel, Düren, Germany), and sequenced at the Eurofins sequencing service (France). The obtained sequences were compared to those obtain from bacterial culture in ISE6 cells and then submitted to the BLAST (basic local alignment search tool) platform to search for sequences with similarity to the *A. phagocytophilum* NV2Os strain.

### 4.10. Enzyme Linked Immunosorbent Assay (ELISA)

Mouse blood samples were collected from the retro-orbital sinus with high precision Pasteur pipettes before each immunization (days 0, 14, 28), on the day of infestation (day 42), at day 56, and at the day of euthanasia (day 90). After centrifugation of blood, serum was collected and stored at -20 °C until used.

Sheep blood samples were collected from the jugular vein in a previously disinfected area using 10 mL vacutainer tubes. The samples were obtained before each immunization (days 0, 15, 30), at days 45, 51, and at the time of the euthanasia (day 73). Sera were collected following centrifugation and stored at −20 °C until ELISAs were performed.

Polystyrene microtiter ELISA plates (MaxiSorb, Roskilde, Denmark) were coated with 100 µL per well of 10 µg/mL each peptide with linear amidated forms of either MIP or SIFa (see peptide sections in Material and Methods) diluted in Carbonate-bicarbonate buffer, pH 9.6 (Sigma-Aldrich, Darmstadt, Germany). After all incubations, the plates were washed three times with 250 µL of PBS containing 0.05% Tween 20 (Montanox 20) (Seppic, La Garenne Colombes, France). The plates were then incubated at 37 °C for two hours and washed three times with 250 µl of PBS containing 0.05% Tween 20. The plates were blocked with PBS containing 1% bovine gelatin (Sigma-Aldrich, Darmstadt, Germany) overnight at room temperature. Serial dilutions of serum were added in duplicate. The plates were incubated with 1:3000 rabbit anti-sheep immunoglobulin G (IgG)-horseradish peroxidase conjugate (Thermo Fisher Scientific, Waltham, MA, USA) and 1:6000 goat anti-mouse immunoglobulin G1 (IgG1)-horseradish peroxidase conjugate (Thermo Fisher Scientific, Waltham, MA, USA) for the detection of sheep and mouse antibodies, respectively. The reaction was revealed by adding 3,3′,5,5′-Tetramethylbenzidine (Thermo Fisher Scientific, Waltham, MA, USA) for 5 min and stopped with 50 µL of 2N H2SO4. The optical density (450 nm) was measured in an ELISA reader (Multiskan, Thermo Fisher Scientific, Waltham, MA, USA). The antibody titers were expressed in arbitrary units (AU) by reference to a calibrated standard anti-ovalbumin IgG.

## 5. Conclusions

Based on the results obtained here, we conclude that vaccine-mediated protection against nymphs in mice and larvae on sheep could not be demonstrated in the present study, despite a promising negative correlation between antibody levels of anti-MIP and larva to nymph molting success. Nevertheless, our results attest to the difficulty of evaluating tick vaccine candidates in target species, such as sheep, in which *I. ricinus*-vaccine trials have not previously been performed. Our data also demonstrate that the MAPs approach is a useful tool for evaluation of small peptides as targets for vaccine candidates against tick infestations, as shown by effective generation of antibodies against two neuropeptide molecules. Although our leading hypothesis was that antibodies against SIFa and/or MIP would impair tick feeding through interference with their secretory/excretory mechanisms, we do not know as yet whether the vaccine-elicited antibodies absorbed by ticks were actually capable of targeting and neutralizing the neuropeptides’ physiological functions. It should also be noted that although the roles of MIP and SIFa in regulation of tick salivary gland and hindgut function are well established, whether disruption of their activities would affect tick physiology is at present unknown. Finally, the use of MAP technology in the present study was validated as a simple and cost-effective method for eliciting antibodies against tick peptide epitopes of interest, offering a valuable strategy to support functional analyses and antigen validation. Ticks have existed for hundreds of millions of years and creating tools that outsmart them is not an easy task. Unfortunately, this is another example showing that the generation of specific antibodies is not a guarantee for protection in the development of vaccines against ticks, and it would probably be interesting in the future to evaluate not only humoral but also cellular immune responses elicited by vaccinal candidates against these highly evolved vectors.

## Figures and Tables

**Figure 1 pathogens-09-00900-f001:**
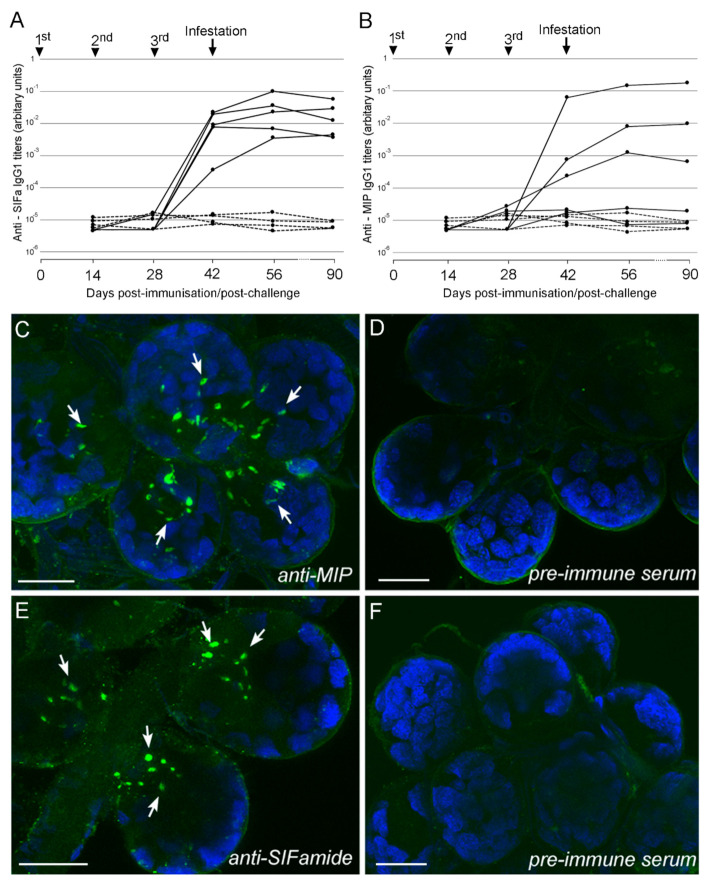
Antibody (IgG1) response to SIFamide (SIFa; (**A**)) and Myoinhibitory (MIP; (**B**)) peptides in vaccinated mice and whole-mount immunohistochemistry on the salivary glands of an *Ixodes ricinus* unfed female (**C**–**F**). Antibody titers were determined by ELISA in serum samples collected at different time points from day 0 to day 90 against the specific peptide both in vaccinated mice (plain lines) and control mice (dashed lines) that received only adjuvant, and represented as arbitrary units (AU). Arrows indicate dates for 1st, 2nd, 3rd immunizations (days 0, 14 and 28) and tick infestations (Day 42). (C) and (E) show the reaction of anti-mouse antibody generated by MIP or SIFa multiple antigen peptide (MAP) constructs, respectively. Note that both antibodies recognized specific axon terminals (green, arrows) in acini type II and III of salivary glands. (D) and (F) are the negative controls where only pre-immune serum of mice subsequently immunized by MIP or SIFa was used, respectively. Note that no reaction in axon terminals in salivary gland acini was observed when pre-immune sera were used. Bar is 10 μm.

**Figure 2 pathogens-09-00900-f002:**
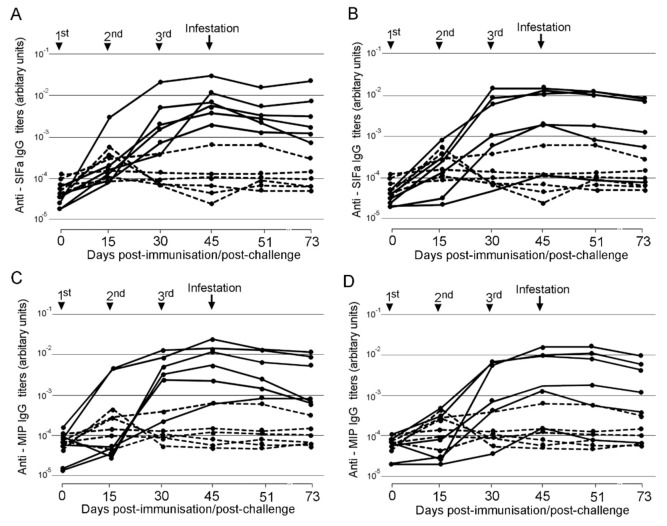
Antibody (IgG) response to SIFamide (SIFa) (**A**,**B**) and Myoinhibitory (MIP) (**C**,**D**) peptides in sheep vaccinated with SIFa (**A**), MIP (**C**), both SIFa and MIP (**B**,**D**) (plain lines), and in control injected with adjuvant (dashed lines). Antibody titers were determined by ELISA in serum samples collected at different time points from day 0 to day 73 against the specific peptide and are represented as arbitrary units (AU). Arrows indicate dates for 1st, 2nd, and 3rd immunizations (Days 0, 15, and 30) and tick infestations (Days 45, 46).

**Figure 3 pathogens-09-00900-f003:**
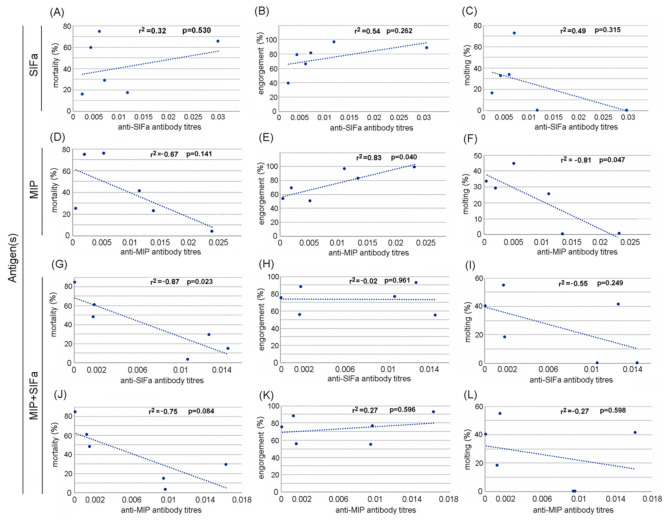
Correlation between antibody titers against SIFamide (SIFa) (**A**–**C**, **G**–**I**) and myoinhibitory (MIP) (**D**–**F**, **J**–**L**) neuropeptides in sheep vaccinated against SIFa (**A**–**C**), MIP (**D**–**F**), both SIFa and MIP (**G**–**L**), and *I. ricinus* larvae mortality, engorgement and molting at Day 45 after immunization. The linear correlation coefficients (R2) and *p*-values are shown (N = 6). Antibody titers are represented as arbitrary units.

**Figure 4 pathogens-09-00900-f004:**
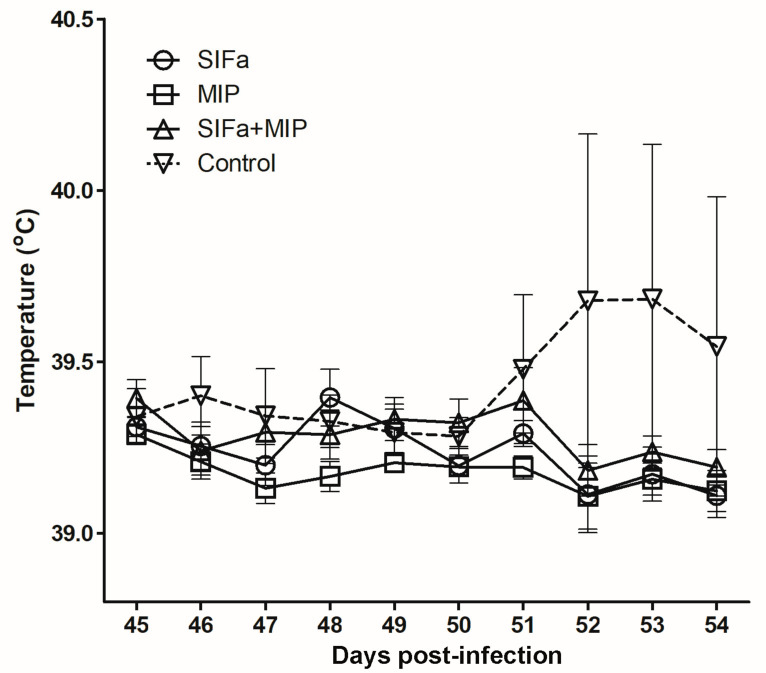
Daily temperature (°C) recorded for groups of sheep vaccinated with SIFamide (SIFa), Myoinhibitory (MIP), and both SIFamide and Myoinhibitory (SIFa + MIP) peptides or injected with adjuvant (Control) from 45 to 54 days after immunisation. Animals were infested with *Anaplasma phagocytophilum*-infected *Ixodes ricinus* nymphs on day 46. Note that the dotted line representing the control group has a very large standard deviation due to a single sheep with a high fever of up to 42 °C on day 52. Results are presented as means +/− standard deviation.

**Figure 5 pathogens-09-00900-f005:**
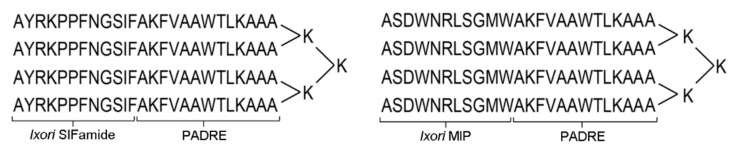
Schematic depiction of multiple antigen peptides (MAPs) used in this study. Note that *Ixodes ricinus* mature neuropeptide B-cell epitops (SIFa or MIP) were fused to a T-helper cell epitop PADRE to construct 4-branched MAPs.

**Table 1 pathogens-09-00900-t001:** Effect of vaccination of mice against SIFamide (SIF) and Myoinhibitory peptide (MIP) on tick infestation parameters.

Tick Parameters	Fed on Control Mice	Fed on SIF Vaccinated Mice	Fed on MIP Vaccinated Mice
Engorgement (No of engorged nymphs/No of nymphs used for infestation)	18/20	20/20	18/20
	3/20	20/20	9/20
	3/20	15/20	18/20
	18/20	16/20	15/20
		17/20	9/20
Total	42/80	88/100 *	69/100
Group average	53%	88%	69%
Feeding (mean weight after feeding/engorged nymph) (mg)	4.37 (±1.35)	4.00 (±1.33)	3.89 (±1.13) *
Molting (No adults at day 109/No engorged nymphs)	8/18	17/20	8/18
	3/3	14/20	7/9
	1/3	15/15	18/18
	16/18	16/16	8/15
		15/17	7/9
Total	28/42	77/88 e *	48/69
Group average	67%	89%	71%
Tick mortality (No of dead nymphs at day 109/total No engorged nymphs)	10/18	3/20	10/18
	0/3	6/20	2/9
	2/3	0/15	0/18
	2/18	0/16	7/15
		2/17	2/9
Total	14/42	11/88 *	21/69
Group average	33.5%	11%	29.4%

Results are shown for each mouse of groups of 5 for SIF- and MIP-vaccinated mice, and 4 mice for the control group. Data were analyzed statistically to compare results between nymphs fed on vaccinated and control mice by student’s *t*-test (* *p* < 0.05), *χ*^2^-test (* *p* < 0.05, Total) at the global level for each group and at the individual level by ANOVA analysis (*p* < 0.05, Group average).

**Table 2 pathogens-09-00900-t002:** Effect of vaccination of sheep with SIFamide (SIFa) and Myoinhibitory (MIP) peptides on tick infestation parameters.

Tick Parameters	Fed on Control Sheep	Fed on SIFa Vaccinated Sheep	Fed on MIP Vaccinated Sheep	Fed on SIFa + MIP Vaccinated Sheep
Larvae				
Engorgement (No of engorged larvae/No of larvae used for infestation)	1340/2067	1483/1836	871/1747	1377/1836
	1499/1701	1309/2000	1454/2126	1561/1789
	49/139	45/117	526/548	144/156
	156/196	242/308	56/105	21/38
	205/376	277/316	1027/1040	123/226
	207/553	863/900	187/228	437/576
Total	3456/5032	4219/5477 **	4121/5794 *	3663/4621 **
Group average	60%	74%	75%	73%
Molting (No nymphs at Day 90/No alive engorged larvae)	31/144	769/1063	94/212	87/217
	54/217	112/339	108/373	112/616
	14/37	6/38	79/312	42/102
	39/128	32/99	14/42	6/11
	0/166	0/97	3/991	0/105
	0/169	0/722	0/145	0/424
Total	138/861	919/2358 **	298/2075	247/1475
Group average	19%	26%	22%	26%
Tick mortality (No of dead ticks at Day 90/total No engorged larvae)	1196/1340	420/1483	659/871	1160/1377
	1282/1499	970/1309	1081/1454	945/1561
	12/49	7/45	214/526	42/144
	28/156	143/242	14/56	10/21
	39/205	180/277	36/1027	18/123
	38/207	141/863	42/187	13/437
Total	2595/3456	1738/4219 **	2046/4121 **	2188/3663 **
Group average	42%	43%	40%	40%
Nymphs				
Engorgement (No of engorged nymphs/No of nymphs used for infestation)	10/48	0/48	1/48	0/48
	1/48	1/48	3/48	6/48
	0/48	0/48	0/48	1/48
	0/48	0/48	1/48	1/48
	0/48	0/48	0/48	0/48
	0/48	0/48	2/48	0/48
Total	11/288	1/288 **	7/288	8/288
Group average	3.8%	0.3%	2.4%	2.8%
Molting (No adults at Day 90/No engorged nymphs)	6/10	0/0	0/1	0/0
	1/1	1/1	3/3	5/6
	0/0	0/0	0/0	0/1
	0/0	0/0	0/1	0/1
	0/0	0/0	0/0	0/0
	0/0	0/0	1/2	0/0
Total	7/11	1/1	4/7	5/8
Group average	80%	100%	37.5%	27.8%
Tick mortality (No of dead ticks at Day 90/total No attached nymphs)	35/41	36/36	27/27	34/34
	34/35	28/29	36/39	31/36
	25/25	31/31	27/27	47/48
	30/30	39/39	29/29	42/43
	39/39	41/41	48/48	40/40
	48/48	45/45	41/42	46/48
Total	211/218	220/221*	218/222	240/249
Group average	97%	99%	98%	96%

Results are shown for each sheep of groups of 6 sheep analyzed for each condition. Data were analyzed statistically to compare results between ticks fed on vaccinated and control sheep at the global level for each group by *χ*^2^-test (* *p* < 0.05, ** *p* < 0.005, Total), and at the individual level by ANOVA analysis (*p* < 0.05, Group average).

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
