# Peer review of "Multiple Antigenic Peptide-Based Vaccines Targeting Ixodes ricinus Neuropeptides Induce a Specific Antibody Response but Do Not Impact Tick Infestation"

_pathogens, 2020, doi:10.3390/pathogens9110900_

Round 1
Reviewer 1 Report
I think that this is a very interesting paper.
In Materials and Methods, the description of the statistical analysis is missing.
Author Response
I think that this is a very interesting paper.
Thanks for your positive comment
In Materials and Methods, the description of the statistical analysis is missing.
The statistical analysis is described in Material and Methods lines 338-344 for mice:
“Tick engorgement (No of engorged nymphs/No of nymphs used for infestation), mortality (No of dead nymphs at day 109/total No engorged nymphs), feeding (mean weight after feeding/engorged nymph) and molting (No adults at day 109/No engorged nymphs) were evaluated and compared between vaccinated mice and controls by student’s t-test with unequal variance (p=0.05) and X2-test (p=0.05). A correlation analysis was conducted in Microsoft Excel (Version 16.16.19) to compare the effect of vaccination on engorgement, mortality, feeding and molting after feeding on vaccinated or control mice with antibody titers at day 42, when tick infestation was performed.”
And lines 380-385 for sheep:
“For both larvae and nymphs, tick engorgement (No of engorged tick/No of tick used for infestation), mortality (No of dead tick at day 90/total No engorged-attached tick), and molting (No nymphs-adults at day 90/No alive engorged larvae-nymphs at day 90) were evaluated and compared between vaccinated sheep and controls by X2-test (p=0,05). A correlation analysis to compare the effect of vaccination on tick engorgement, mortality, and molting, after feeding on vaccinated or control sheep, with antibody titers at day 45, was conducted in Microsoft Excel (Version 16.16.19)." In the new version, we added the fact that analysis was performed via “Two-tailed Pearson’s correlation analysis”
Reviewer 2 Report
This manuscript describes the evaluation of an experimental vaccine against Ixodes ricinus ticks. The authors used two different experimental models; nymph infestation of immunized mice and larval and Anaplasma phagocytophilum-infected nymph infestation of sheep. The experimental vaccine preparations were made of multi-antigenic peptides (MAPs) that contained epitopes of a recombinant tick neuropeptide conjugated to a T-helper epitope. Two different neuropeptides were used, SIFa and MIP. which play a role in salivary gland and hindgut activity, respectively. These studies fall within the scope of the journal.
Immunization with different experimental vaccines induced antibodies in mice and sheep. Results from infestation of mice revealed that the number of nymphs that fed on SIFa immunized mice was statistically significant higher than that of the other groups of mice (MIP vaccinated or adjuvant only group). This was also reflected in higher molting rate and decreased tick mortality. In the sheep model, essentially the same result was found when SIFa immunized sheep were infested with larvae. However, an opposite effect was found when sheep on the nymphs that were used for infestation; engorgement rate was decreased, and mortality was very lightly increased.
The authors have done the analysis of the tick parameters on the total number of ticks per experimental group. That is not correct. Analysis should be done per experimental animal, and from these values a group average can be calculated.
The evaluation of the effect of treatment on the transmission of Anaplasma phagocytophilum to sheep could not be completed because of the fact that only one out six control sheep showed evidence of A. phagocytophilum infection.
More detailed comments are provided in the section below.
Introduction
Line 68-69 The fact that the number of adult female ticks that have fully engorged is reduced on cattle that have been immunized with Bm86-based vaccine, suggests that also earlier stages than the adult female stage is affected. This is further corroborated by the fact that in vitro experiments with Rhipicephalus microplus larvae revealed that feeding was limited when larvae were fed serum from cattle that has been immunized against Bm86.
Results
In the Materials and Methods section, it is stated that the control groups of mice that received adjuvant only, comprised only 4 mice instead of five because one of the mice had unexpectedly died (Lines 329-331). Yet, in figure 1A and 1B the results of five control mice are presented, and according to the legends of figure 1, these control mice had received only adjuvant (Lines 107-109). The authors are requested to comment on this.
Results from infestation of mice surprisingly revealed that the total number of nymphs that fed on SIFa immunized mice was statistically significant higher than that of the other groups of mice (MIP vaccinated or adjuvant only group). This was also reflected in higher molting rate and decreased tick mortality. The authors state that there was no correlation between antibody levels and tick feeding and development parameters (data not shown; Lines 145-147). To allow such analysis, it is of importance to know what the tick numbers for each mouse were, because presently it is not known whether a decrease in the total number of engorged ticks was due to the fact that engorgement failed in one animal. These data should be added.
There was no effect of immunization on tick mortality when using nymph infestation of sheep. The authors claim that there is a statistical effect on tick mortality, but that seems to be statistical shopping. When the X2 is used including all experimental groups, there is no statistically significant difference. Only when the groups are analysed separately with the control group, the group of ticks that had fed on SIFa vaccinated sheep showed a difference with the ticks that had fed on control sheep (p=0.03, Pearsons). The authors are requested to comment on this.
Table 2: The percentage molting 138/861 (No nymphs at Day 90/No alive engorged larvae) should read 16% and not 1% (this must be corrected).
The authors state that in addition to a negative impact on tick engorgement (though the difference was statistically significant only in SIFa-vaccinated sheep), there was also an effect on molting (Lines 195-199). This is not corroborated by the data, and this statement should be removed.
There is a discrepancy when the group averages of the tick parameters are deduced from the figure 3, and those reported in table 2. That is most likely due to the fact that in table 2 the authors have made calculations based on the total values. That is not correct. See table below for the deduced values from the figures.
|
MORTALITY |
SIFa |
MIP |
SIFa+MIP |
|
75 |
75 |
85 |
|
|
66 |
76 |
61 |
|
|
60 |
41 |
49 |
|
|
30 |
26 |
30 |
|
|
18 |
22 |
15 |
|
|
2 |
3 |
2 |
|
|
Average |
41,8 |
40,5 |
40,3 |
|
ENGORGEMENT |
SIFa |
MIP |
SIFa+MIP |
|
98 |
100 |
92 |
|
|
90 |
98 |
88 |
|
|
81 |
82 |
76 |
|
|
80 |
70 |
75 |
|
|
70 |
58 |
57 |
|
|
39 |
52 |
56 |
|
|
Average |
76,3 |
76,7 |
74,0 |
|
MOLTING |
SIFa |
MIP |
SIFa+MIP |
|
72 |
45 |
55 |
|
|
34 |
34 |
41 |
|
|
32 |
29 |
40 |
|
|
18 |
26 |
19 |
|
|
1 |
1 |
1 |
|
|
0 |
1 |
1 |
|
|
Average |
26,2 |
22,7 |
26,2 |
The analysis should be done per sheep, and from these values a group average can be calculated. This should be corrected.
The authors present a correlation analysis of specific antibody response and different tick parameters in figure 3. It is advised to add the p-value to the graphs, which informs the reader about the statistical significance of the R2 value.
Figure 2F, remove the double R2 value entry at the bottom.
Author Response
This manuscript describes the evaluation of an experimental vaccine against Ixodes ricinus ticks. The authors used two different experimental models; nymph infestation of immunized mice and larval and Anaplasma phagocytophilum-infected nymph infestation of sheep. The experimental vaccine preparations were made of multi-antigenic peptides (MAPs) that contained epitopes of a recombinant tick neuropeptide conjugated to a T-helper epitope. Two different neuropeptides were used, SIFa and MIP. which play a role in salivary gland and hindgut activity, respectively. These studies fall within the scope of the journal.
Immunization with different experimental vaccines induced antibodies in mice and sheep. Results from infestation of mice revealed that the number of nymphs that fed on SIFa immunized mice was statistically significant higher than that of the other groups of mice (MIP vaccinated or adjuvant only group). This was also reflected in higher molting rate and decreased tick mortality. In the sheep model, essentially the same result was found when SIFa immunized sheep were infested with larvae. However, an opposite effect was found when sheep on the nymphs that were used for infestation; engorgement rate was decreased, and mortality was very lightly increased.
The authors have done the analysis of the tick parameters on the total number of ticks per experimental group. That is not correct. Analysis should be done per experimental animal, and from these values a group average can be calculated.
Thank you for that constructive comment. As requested, the results are now presented by animal in the revised version (see below)
The evaluation of the effect of treatment on the transmission of Anaplasma phagocytophilum to sheep could not be completed because of the fact that only one out six control sheep showed evidence of A. phagocytophilum infection.
More detailed comments are provided in the section below.
Introduction
Line 68-69 The fact that the number of adult female ticks that have fully engorged is reduced on cattle that have been immunized with Bm86-based vaccine, suggests that also earlier stages than the adult female stage is affected. This is further corroborated by the fact that in vitro experiments with Rhipicephalus microplus larvae revealed that feeding was limited when larvae were fed serum from cattle that has been immunized against Bm86.
We agree with this comment as, for this one-host tick, the impact is evaluated on females recovered after 21-24 days of larvae infestation, but of course, larvae and nymphs are also affected by antibodies taken with the blood meal. Accordingly, the sentence was modified as follows in the new version: “The effect of recombinant Bm86 vaccine is based on reduction in tick infestation due to a diminished capacity of ticks to feed and – for the females – to subsequently reproduce”
Results
In the Materials and Methods section, it is stated that the control groups of mice that received adjuvant only, comprised only 4 mice instead of five because one of the mice had unexpectedly died (Lines 329-331). Yet, in figure 1A and 1B the results of five control mice are presented, and according to the legends of figure 1, these control mice had received only adjuvant (Lines 107-109). The authors are requested to comment on this.
We are very sorry about this error due to a data inversion. There were indeed only 4 mice in the control group and this error was corrected in the new version.
Results from infestation of mice surprisingly revealed that the total number of nymphs that fed on SIFa immunized mice was statistically significant higher than that of the other groups of mice (MIP vaccinated or adjuvant only group). This was also reflected in higher molting rate and decreased tick mortality. The authors state that there was no correlation between antibody levels and tick feeding and development parameters (data not shown; Lines 145-147). To allow such analysis, it is of importance to know what the tick numbers for each mouse were, because presently it is not known whether a decrease in the total number of engorged ticks was due to the fact that engorgement failed in one animal. These data should be added.
As requested, the table 1 was modified and results are now presented for each mouse individually
There was no effect of immunization on tick mortality when using nymph infestation of sheep. The authors claim that there is a statistical effect on tick mortality, but that seems to be statistical shopping. When the X2 is used including all experimental groups, there is no statistically significant difference. Only when the groups are analysed separately with the control group, the group of ticks that had fed on SIFa vaccinated sheep showed a difference with the ticks that had fed on control sheep (p=0.03, Pearsons). The authors are requested to comment on this.
We agree with the fact that the difference is only significant between SIFa and control group and the new text version was modified in order to avoid any confusion.
Table 2: The percentage molting 138/861 (No nymphs at Day 90/No alive engorged larvae) should read 16% and not 1% (this must be corrected).
This was corrected in the new version
The authors state that in addition to a negative impact on tick engorgement (though the difference was statistically significant only in SIFa-vaccinated sheep), there was also an effect on molting (Lines 195-199). This is not corroborated by the data, and this statement should be removed.
This was modified in the new version
There is a discrepancy when the group averages of the tick parameters are deduced from the figure 3, and those reported in table 2. That is most likely due to the fact that in table 2 the authors have made calculations based on the total values. That is not correct. See table below for the deduced values from the figures.
|
MORTALITY |
SIFa |
MIP |
SIFa+MIP |
|
|
75 |
75 |
85 |
|
|
66 |
76 |
61 |
|
|
60 |
41 |
49 |
|
|
30 |
26 |
30 |
|
|
18 |
22 |
15 |
|
|
2 |
3 |
2 |
|
|
|
|
|
|
Average |
41,8 |
40,5 |
40,3 |
|
|
|
|
|
|
|
|
|
|
|
ENGORGEMENT |
SIFa |
MIP |
SIFa+MIP |
|
|
98 |
100 |
92 |
|
|
90 |
98 |
88 |
|
|
81 |
82 |
76 |
|
|
80 |
70 |
75 |
|
|
70 |
58 |
57 |
|
|
39 |
52 |
56 |
|
|
|
|
|
|
Average |
76,3 |
76,7 |
74,0 |
|
|
|
|
|
|
|
|
|
|
|
MOLTING |
SIFa |
MIP |
SIFa+MIP |
|
|
72 |
45 |
55 |
|
|
34 |
34 |
41 |
|
|
32 |
29 |
40 |
|
|
18 |
26 |
19 |
|
|
1 |
1 |
1 |
|
|
0 |
1 |
1 |
|
|
|
|
|
|
Average |
26,2 |
22,7 |
26,2 |
The analysis should be done per sheep, and from these values a group average can be calculated. This should be corrected.
As requested the table 2 was modified in order to show results at the individual level and group means calculated with each animal mean.
The authors present a correlation analysis of specific antibody response and different tick parameters in figure 3. It is advised to add the p-value to the graphs, which informs the reader about the statistical significance of the R2 value.
The p-values obtained for the two-tailed Pearson’s correlation analysis were added in the new version of the figure 3.
Figure 2F, remove the double R2 value entry at the bottom.
This was corrected in the new version
Reviewer 3 Report
General Comments
The manuscript by Almazan and colleagues entitled "Multiple antigenic peptide-based vaccines targeting Ixodes ricinus neuropeptides induce a specific antibody response but do not impact tick infestation" examines the use tick neuropeptides to produce vaccines that protect against tick infestation in mice and sheep. The authors have experience with efforts to produce vaccines based on tick proteins, and like almost all previous efforts this one appears to have failed. Although publication of a negative study is laudable, the result should prompt rethinking of this logical but difficult approach to prevention of tickborne diseases.
Major Comments
1. The Introduction provides an excellent overview of the rationale for attempts to produce tick-directed vaccines. Unfortunately almost all of these attempts have failed, and although the current study utilizes a more refined peptide immunization system, the outcome is disappointing. The authors try to spin the tick molting interference as best they can, but this questionable result is far less important than protection against tick feeding and consequent disease transmission.
2. What can we learn from this negative study? For one thing, antibody response to tick proteins may be less important than cellular responses induced by tick vaccines. The authors should explore systems to examine cellular immunity related to tick peptides for future vaccine studies.
3. Another lesson pertains to the changing clinical picture of tickborne diseases. These diseases are much more complex than previously thought, and we now have Relapsing Fever Borrelias, viruses, fungi, nematodes and various Bartonella species in the tickborne soup. It is clear that both hard and soft ticks play a role in disease transmission to humans, so a tick vaccine should protect against both of these vectors (Shah et al. Healthcare 2019;7:121; Fesler et al. Healthcare 2020;8:97). It follows that a successful vaccine should target antigens that are present in both hard and soft ticks. An example of this broad target are the lipocalins found in hard and soft tick salivary glands, and other shared targets probably exist (Mans et al. Mol. Biol. Evol. 2003;20:1158; Maldonado-Ruiz et al. Front. Immunol. 2019:10:1996; Chmelaˇr et al. Front. Physiol. 2019;10:812). These proteins may work better from a clinical perspective to prevent transmission of the broad range of tickborne diseases.
Minor Comments
1. Line 398: "special"
2. Line 444: "tick vaccine candidates"
Author Response
General Comments
The manuscript by Almazan and colleagues entitled "Multiple antigenic peptide-based vaccines targeting Ixodes ricinus neuropeptides induce a specific antibody response but do not impact tick infestation" examines the use tick neuropeptides to produce vaccines that protect against tick infestation in mice and sheep. The authors have experience with efforts to produce vaccines based on tick proteins, and like almost all previous efforts this one appears to have failed. Although publication of a negative study is laudable, the result should prompt rethinking of this logical but difficult approach to prevention of tickborne diseases.
Thank you for your positive and encouraging comments, as well as for your constructive suggestions
Major Comments
- The Introduction provides an excellent overview of the rationale for attempts to produce tick-directed vaccines. Unfortunately almost all of these attempts have failed, and although the current study utilizes a more refined peptide immunization system, the outcome is disappointing. The authors try to spin the tick molting interference as best they can, but this questionable result is far less important than protection against tick feeding and consequent disease transmission.
We fully agree with the reviewer and the discussion has been slightly modified to this effect in the hope that the new version is more appropriate
- What can we learn from this negative study? For one thing, antibody response to tick proteins may be less important than cellular responses induced by tick vaccines. The authors should explore systems to examine cellular immunity related to tick peptides for future vaccine studies.
This is sound advice for which we thank you, and the following sentence was added at the end of the conclusion: “Unfortunately, this is another example showing that the generation of specific antibodies is not a guarantee for protection in the development of vaccines against ticks, and it would probably be interesting in the future to evaluate not only humoral but also cellular immune responses elicited by vaccinal candidates against these highly evolved vectors.”
- Another lesson pertains to the changing clinical picture of tickborne diseases. These diseases are much more complex than previously thought, and we now have Relapsing Fever Borrelias, viruses, fungi, nematodes and various Bartonella species in the tickborne soup. It is clear that both hard and soft ticks play a role in disease transmission to humans, so a tick vaccine should protect against both of these vectors (Shah et al. Healthcare 2019;7:121; Fesler et al. Healthcare 2020;8:97). It follows that a successful vaccine should target antigens that are present in both hard and soft ticks. An example of this broad target are the lipocalins found in hard and soft tick salivary glands, and other shared targets probably exist (Mans et al. Mol. Biol. Evol. 2003;20:1158; Maldonado-Ruiz et al. Front. Immunol. 2019:10:1996; Chmelaˇr et al. Front. Physiol. 2019;10:812). These proteins may work better from a clinical perspective to prevent transmission of the broad range of tickborne diseases.
We fully share this view on the need to identify common targets for several species of ticks, whether hard or soft. And this is one of the reasons why we turned to these neuropeptides which, as indicated in line 84-86 of the introduction, are most likely common to several species of ticks.
Minor Comments
- Line 398: "special"
This was corrected
- Line 444: "tick vaccine candidates"
This was corrected
Round 2
Reviewer 2 Report
This is a revised version of a earlier.
The authors have presented the more detailed data per animal as requested. Data are listed in tables 1 and 2. Upon evaluation of these tables, there are a number of mistaken/anomalies that need to be corrected (see attached file). For instance, in table 1 (mouse experiment) the numbers of dead ticks (nymphs) and number of nymphs that molted to adults, do not add up to the number of engorged nymphs.
Similar mistakes/anomalies are found in table 2 (sheep experiment), both for the results obtained with larvae and with nymphs. For instance, in the case of the nymphs, the data shows that 58 out of 58 nymphs attached whereas only 48 nymphs were applied to the animal. Also, bottom page 6 the authors state that "Out of a total of 1352 nymphs used to
infest sheep....". The number of nymphs used was actually 1152 and not 1352.
The reason to ask for the per animals information was because the statistical unit in these experiments is the experimental animal. Hence, to calculate effects, the parameters (engorgement, molting, mortality) must be first calculated per individual animal, and the difference between the different groups can subsequently be calculated. None of the parasitological parameters appeared to be statistically significant different between experimental groups (see attached file).
The authors calculate the average percentage of nymphs that molt to adults after having been fed on sheep. In cases no engorged nymphs were retrieved from the sheep, the authors include such cases as 0% molting to adults. This is scientifically not correct; the percentage of molting musty be calculated from the number of nymphs that were incubated to molting.
The authors were also requested to present the p-values of the correlation analysis that is presented in figure 3. None of the correlations that are presented reached statistical significance (all p-values are well above 0.05, the lowest p-value reaching 0.14), contrary what the authors state: "For larvae, a statistically significant increase in the engorgement rate was observed in vaccinated sheep either with SIFa or MIP alone or with the combination of the two peptides".
In conclusion, this manuscript needs to be checked more thoroughly for inconsistencies, typo's and other mistakes in the data that are presented, especially in the tables 1 and 2.
The statistical analysis is not correct. The statistical unit is the individual animal, and these data should be used to compare the average values per group.
The Results and Discussion section need to be rewritten taking into account the comments provided above. All statements must be corroborated by the data that are provided.
The fact that this is a 'negative' experiment does not exclude this manuscript from publication.

Author Response
This is a revised version of a earlier.
We warmly thank the reviewer for his/her very precise analysis of all the data presented and for his/her help in the statistical analysis of the results. We are sorry for the errors in the 2 results tables and Figure 3 that have now been corrected, and we believe that everything is okay now. The statistical analysis carried out by the reviewer (ANOVA) has now been used in the new version and the text adapted according to the results obtained. However, we have also retained the X2-test analysis that compares groups as it is the one used in most published studies evaluating anti-tick vaccine candidates.
The authors have presented the more detailed data per animal as requested. Data are listed in tables 1 and 2. Upon evaluation of these tables, there are a number of mistaken/anomalies that need to be corrected (see attached file). For instance, in table 1 (mouse experiment) the numbers of dead ticks (nymphs) and number of nymphs that molted to adults, do not add up to the number of engorged nymphs.
These mistakes were corrected in the new version.
Similar mistakes/anomalies are found in table 2 (sheep experiment), both for the results obtained with larvae and with nymphs. For instance, in the case of the nymphs, the data shows that 58 out of 58 nymphs attached whereas only 48 nymphs were applied to the animal. Also, bottom page 6 the authors state that "Out of a total of 1352 nymphs used to
infest sheep....". The number of nymphs used was actually 1152 and not 1352.
These mistakes were corrected in the new version. However, regarding nymphs, we want to highlight that the mortality was calculated according to attached nymphs and not to engorged nymphs because of the very low percentage of engorged nymphs. Indeed, regarding the “False” mention of the attached excel file provided by the reviewer, for the first control sheep for example, we have 35 attached dead ticks (B99) + 6 that molted (B79)– 41 attached ticks (B88) =0.
The reason to ask for the per animals information was because the statistical unit in these experiments is the experimental animal. Hence, to calculate effects, the parameters (engorgement, molting, mortality) must be first calculated per individual animal, and the difference between the different groups can subsequently be calculated. None of the parasitological parameters appeared to be statistically significant different between experimental groups (see attached file).
This was modified in the new version
The authors calculate the average percentage of nymphs that molt to adults after having been fed on sheep. In cases no engorged nymphs were retrieved from the sheep, the authors include such cases as 0% molting to adults. This is scientifically not correct; the percentage of molting musty be calculated from the number of nymphs that were incubated to molting.
This was modified in the new version
The authors were also requested to present the p-values of the correlation analysis that is presented in figure 3. None of the correlations that are presented reached statistical significance (all p-values are well above 0.05, the lowest p-value reaching 0.14), contrary what the authors state: "For larvae, a statistically significant increase in the engorgement rate was observed in vaccinated sheep either with SIFa or MIP alone or with the combination of the two peptides".
This last result did not come from the correlation analysis but from the X2-test analysis done on the data per group. The results are now presented separately and we hope this will be satisfactory for the reviewer. In addition, and for this we apologize, there were some mistakes in the p-values of the figure 3 that are now corrected and the degree of statistical significance is indicated both in the figure 3 and in the text.
In conclusion, this manuscript needs to be checked more thoroughly for inconsistencies, typo's and other mistakes in the data that are presented, especially in the tables 1 and 2.
We hope that no error remains
The statistical analysis is not correct. The statistical unit is the individual animal, and these data should be used to compare the average values per group.
This was changed as requested by the reviewer
The Results and Discussion section need to be rewritten taking into account the comments provided above. All statements must be corroborated by the data that are provided.
This was modified as requested by the reviewer
The fact that this is a 'negative' experiment does not exclude this manuscript from publication.
Round 3
Reviewer 2 Report
This is a revised version of an earlier manuscript.
The authors have made adequate changes to the manuscript.